# The Direct and Indirect Effects of Tyrosine Kinase Inhibitors on the Cardiovascular System in Chronic Myeloid Leukemia

Alessandro Costa [1], Raimondo Pittorru [2], Giovanni Caocci [1], Federico Migliore [2], Francesco Tona [2], Olga Mulas [1,*] and Giorgio La Nasa [1]

1 Hematology Unit, Businco Hospital, Department of Medical Sciences and Public Health, University of Cagliari, 09121 Cagliari, Italy; alessandrocosta161195@gmail.com (A.C.); giovanni.caocci@unica.it (G.C.); giorgio.lan@unica.it (G.L.N.)
2 Department of Cardiac, Thoracic, Vascular and Public Health, University of Padua, 35100 Padua, Italy; raipittov93@gmail.com (R.P.); federico.migliore@unipd.it (F.M.); francesco.tona@unipd.it (F.T.)
* Correspondence: mulasolga@unica.it

**Abstract:** Since their introduction, tyrosine kinase inhibitors (TKIs) have radically changed the treatment paradigm of Chronic Myeloid Leukemia (CML), leading to deep and lasting molecular responses and profoundly influencing survival. However, cancer-therapy-related Cardiovascular Toxicities (CTR-CVTs) associated with BCR::ABL1 TKIs are one of the main sources of concern: hypertension, arterial occlusive events, arrhythmias, dysmetabolic alteration, and glomerular filtration impairment are frequently reported in clinical trials and real-life experiences. Therefore, a close interaction between hematologists and cardiologists becomes crucial to implementing prevention protocols based on a comprehensive assessment of baseline cardiovascular risk, the management of any detectable and modifiable risk factors, and the elaboration of a monitoring plan for CTR-CVTs during treatment. Here, we provide the most comprehensive and recent evidence in the literature on the pathophysiological patterns underlying CTR-CVTs, providing useful evidence-based guidance on the prevention and management of CVD risk factors at baseline and during treatment with BCR::ABL1 TKIs.

**Keywords:** TKI; cardiovascular toxicity; CML





## 1. Introduction

The advent of target therapy with tyrosine kinase inhibitors (TKIs) has completely revolutionized the Chronic Myeloid Leukemia (CML) treatment paradigm, leading to a survival equal to that of the general population and causing patients to die mainly from causes unrelated to CML [1,2]. Despite their well-recognized efficacy and excellent long-term survival outcomes in responsive patients, cancer-therapy-related Cardiovascular Toxicities (CTR-CVTs) associated with BCR::ABL1 TKIs are still a source of concern. Endothelial damage, the promotion of atherogenesis, metabolic dysfunctions, and glomerular function impairment are the most frequently reported rationales in the literature.

Particularly, the second- and third-generation TKIs (2G/3G-TKIs), nilotinib and ponatinib, have significantly increased the reporting for myocardial infarction (MI), stroke, peripheral arterial occlusive disease (PAOD), and venous thromboembolic events (VTEs) [3–7]. Similar to other Vascular Endothelial Growth Factor Receptor (VEGFR) inhibitors, ponatinib can cause new-onset arterial hypertension [8], while dasatinib can determine pulmonary arterial hypertension (PAH) [9]. Among new agents, asciminib [10] and vodobatinib [11,12] have shown a good safety profile in clinical trials, while olverembatinib, a third-generation TKI drawn on the ponatinib scaffold, induced arterial hypertension, pericardial effusion, and ventricular extrasystoles in 32.1% of the patients included in phase I/II trials [13]. In this scenario, we identify two major issues: (1) CTR-CVTs may require the premature discontinuation of TKIs, resulting in worsening outcomes; and (2) the burden of cardiovascular

comorbidity may be challenging to manage, requiring great experience, active surveillance, and patient adherence. In order to mitigate the cardiovasc1ular risk at baseline and once therapy is started, the interaction between hematologists and cardiologists becomes crucial. For this purpose, the first Guidelines on Cardio-Oncology were published in 2022 by the European Society of Cardiology (ESC), providing an individualized approach to care based on basal cardiovascular risk assessment and new surveillance protocols during cancer treatment [14]. Moreover, the Cardio-Oncology Study Group of the Heart Failure Association (HFA) of the ESC, in collaboration with the International Cardio-Oncology Society (ICOS), proposed a new scoring tool that can be used specifically to stratify the cardiovascular risk in cancer patients before starting potential cardiotoxic cancer therapies [15].

The aim of this review is to present the most important evidence regarding the underlying mechanism of CTR-CVTs and the cardiovascular safety profiles of BCR::ABL1 TKIs. Accordingly, we try to highlight the preventive management strategies for cardiovascular assessment and risk factor modification before, during, and after TKI treatment.

## 2. Pathogenetic Overview of TKIs-Induced Cardiovascular Events

Through reversible phosphorylation, kinases regulate many signaling pathways that control metabolism, cell cycle progression, cell differentiation, proliferation, and death [16]. All TKIs approved for CML share the ability to inhibit BCR::ABL1, but have also shown heterogeneous off-target toxicities on kinases involved in several physiological processes, such as VEGFR 1 to 3, TIE-3, and platelet-derived growth factor receptors (FGFR) 1 to 4 [17–20].

The off-target toxicities related to TKI treatment and their rationale are summarized in Figure 1.

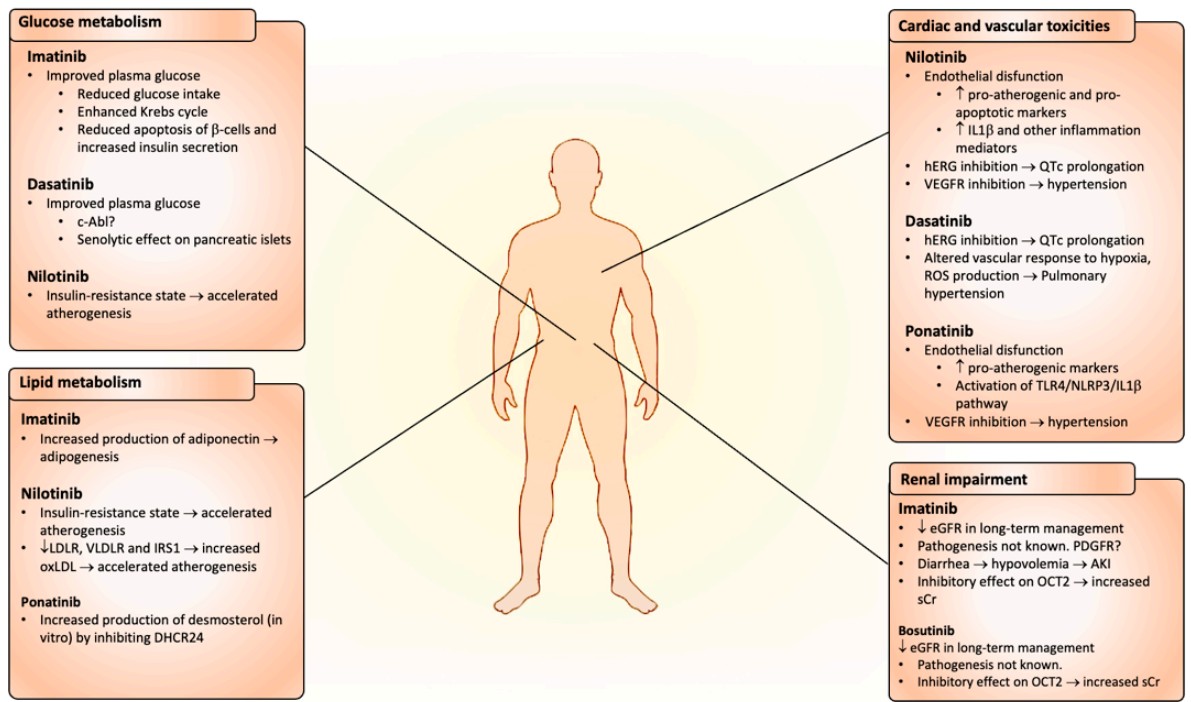

**Figure 1.** Mechanisms underlying off-target toxicities during TKIs treatment. AKI: Acute Kidney Injury; DHCR24: 24-Dehydrocholesterol Reductase; eGFR: estimated Glomerular Filtration Rate; hERG: human Ether-à-go-go-Related Gene; IL1β: Interleukin-1 β; IRS1: Insulin receptor substrate 1; LDLR: Low-density Lipoprotein Receptor; OCT2: Organic Cation Transporter 2; NLRP3: NOD-, LRR-Pyrin Domain-containing 3; oxLDL: oxidized Low-density Lipoprotein; PDGFR: Platelet-derived Growth Factor Receptor; ROS: reactive oxygen species; sCr: serum Creatinine; TLR4: Toll-like Receptor 4; VEGFR: Vascular Endothelial Growth Factor Receptor; and VLDLR: Very-Low-density Lipoprotein Receptor.

*2.1. Atherogenesis and Occlusive Events*

Common hypotheses for adverse occlusive events (AOEs) include pro-atherogenic, anti-angiogenic, and metabolic effects. Initial studies focused on platelet function, driven mainly by the prevalence of arterial vascular events. Different mechanisms seem to be involved at different times of the onset of AOEs: vasospasm [21] or acute thrombotic events can underlie early adverse events, while later events may reflect a chronic inflammatory or pro-thrombotic state [22]. While it is accepted that imatinib, dasatinib, and bosutinib do not influence platelet function towards a prothrombotic state, less clear is the role of nilotinib. In some evidence, nilotinib has not been shown to significantly alter platelet activity [23,24], while others have reported it having an influence on platelet secretion, activation, and adhesion [25]. Indeed, nilotinib has enhanced thrombus growth and stability in human and murine models via ex vivo adhesion on type I collagen under arterial flow and increased in vivo thrombus growth in mouse models [25]. This capacity has been reported recently also for ponatinib [22,25]. The latter has also increased serum P-selectin secretion, known as a marker of platelet and endothelial activation, and other inflammatory markers, including TNF-a, IFN-g, and IL-6 [22].

Other studies have focused on TKIs' influence on the endothelium and atherogenesis. It has been reported that nilotinib induces the expression of pro-atherogenic molecules such as ICAM-1, VCAM1, and E-selectin on endothelial cells (ECs), chemokines such as CCL2, CXCL8, and CXCL2, and pro-apoptotic markers such as caspase-3 and -7 [6,26]. In addition, IL-1b, typically associated with inflammation, atherosclerosis, and type 2 diabetes (T2D), seems to be increased in patients treated with nilotinib [27,28]. Interestingly, a higher frequency of mutation in TET2, ASXL1, and DNMT3A has been noted in CML patients who developed AOEs. Indeed, studies on clonal hematopoiesis have shown an increased risk of cardiovascular disease (CVD) and a greater frequency with age, thus supporting the influence of aging in increasing basal cardiovascular risk and suggesting possible clinical implications in the future [26,29]. Conversely, less is known about the third-generation TKI, ponatinib. Still, it is likely that, like nilotinib, the drug causes the switch from an antithrombotic phenotype to a pro-thrombotic one in ECs. Ponatinib increases the expression of VCAM1 [30], activates the TLR4/NLRP3/IL1-b pathway in cardiac and systemic myeloid cells both in vitro and in vivo [31], and inhibits VEGFR. These abilities may interfere with angiogenesis and increase systemic arterial pressure (BP) [32]. Furthermore, a metabolomic analysis has shown a peculiar metabolic profile in patients treated with nilotinib and ponatinib, such as a higher serum tyrosine level, which is well known to be a marker of systemic inflammation and the impaired functionality of vessel walls [33]. Altogether, these data seem to suggest that nilotinib- and ponatinib-induced atherogenesis and occlusive events depend on their effect on the viability of ECs and on interference with the defensive process from recanalizing and vessel repair once stenosis is established [26,30,31].

*2.2. Hypertension*

The off-target effect of ponatinib and nilotinib includes the inhibition of VEGFR, a tyrosine kinase receptor predominantly expressed on the surface of ECs, which normally regulates endothelial proliferation and survival, improves vascular permeability, and drives angiogenesis. The inhibition of VEGFR is associated with a reduction in nitric oxide (NO) production, a potent vasodilator, and an increase in oxidative stress and endothelin 1, which are associated with inflammation and vasoconstriction [26,32]. Therefore, VEGFR inhibition seems to justify the increased incidence of hypertension in these patients, probably by altering the NO pathway, increasing the production of endothelin-1 and/or inducing microvascular rarefaction [34]. Indeed, NO also participates in tubule-glomerular feedback and sodium metabolism, so its inhibition may contribute to hypertension through sodium retention and another direct renal mechanism. Finally, it is likely that the inflammatory state associated with ponatinib may exacerbate a pre-existing hypertensive condition; according to recent evidence, this could be the result of the polarization of M1 macrophages induced

by an alteration in circulating substrates, such as glucose or lipid, lipotoxicity, and tissue hypoxia [35].

### 2.3. Arrhythmias

Nilotinib and dasatinib have also been found to be potent inhibitors of the human ether-à-go-go-related gene (hERG), also known as KCNH2, which encodes for the pore-forming subunit of the channel-conduction $I_{Kr}$, which is critical for the repolarization of human ventricles [36]. By blocking the efflux of potassium ions, these drugs lengthen the amount of time necessary to regenerate the cardiac action potential, causing changes in the post-contraction refractory period and leading to life-threatening ventricular arrhythmias [37]. Nilotinib has been found to exert inhibitory effects on hERG with an $IC_{50}$ value of 0.13 μM, indicating its ability to prolong the QT interval and potentially induce atrial fibrillation. Furthermore, chronic exposure to nilotinib seems to downregulate the amount of hERG on the cell membrane of human-induced pluripotent stem-cell-derived cardiomyocytes. In the same study, dasatinib demonstrated activity on the same target with an $IC_{50}$ of 14.3 μM, but appeared to rarely cause QT interval prolongation events [37].

### 2.4. Pulmonary Hypertension

Pulmonary hypertension (PH), defined by a mean pulmonary arterial pressure of >20 mmHg at rest [38], is a rare and life-threatening side effect of dasatinib [9]. The exact mechanism is still not entirely clear, but increased vascular resistance in the pulmonary circulation appears to have a multifactorial genesis, including vasoconstriction, the re-modeling of the lung vessel wall, and thrombosis [39]. More specifically, some evidence has shown that the impact of dasatinib on ECs is mainly due to an increased production of mitochondrial ROS, and through attenuated hypoxic pulmonary vasoconstriction responses [40].

### 2.5. Metabolic Effects

Several studies have focused on the metabolic effects induced by TKIs, reporting contrasting results between TKIs. In fact, imatinib has been shown to improve in vitro and in vivo glucose metabolism [41–43] through the internal translocation of glucose transporters (GLUT1, GLUT3, and GLUT6), by reducing glucose intake, switching from glycolysis to the tricarboxylic acid cycle, and increasing insulin secretion by limiting pancreatic b-cell apoptosis [44]. In addition, adiponectin, a hormone secreted by adipocytes with an important effect on insulin sensitivity regulation, has been found to increase in patients treated with imatinib. In this context, imatinib seems to promote the adipogenic differentiation of mesenchymal stromal cells, probably by the inhibition of PDGFRa and PDGFRb, thus interfering in the normal signaling regulation of macrophages on pre-adipocytes and stimulating adipogenesis [45]. With regard to dasatinib, some studies have reported improved plasma glucose levels, probably through its direct effect on c-Abl [46–49] or a senolytic action on pancreatic islets [50–52], while other authors have reported increased glucose levels in patients treated with dasatinib, possibly related to the inhibition of c-KIT, which has proved to be essential for the survival of b-cells in mice [53].

Numerous pieces of evidence support the ability of nilotinib to increase the plasma glucose levels through an insulin resistance (IR) status [54–56]. This condition, characterized by a reduced glucose intake in the skeletal and cardiac muscle, adipose tissue, and liver, has been associated with a low-grade chronic inflammatory state and the production of pro-inflammatory cytokines, including IL-6, IL-8, and TNF-a, thus contributing to EC dysfunction and the acceleration of atherosclerosis [57]. The downregulation of Low-density Lipoprotein Receptor (LDLR), Very-Low-Density Lipoprotein (VLDLR), and Insulin Receptor Substrate 1 (IRS1) and a progressive increase in circulating oxidized LDL levels have also been associated with nilotinib [6,58]. Inhibited lipid internalization by adipose tissue contributes to the IR by increasing free fatty acids (FFA), which are known to

contribute to atherogenesis via macrophage activation, foam cell formation, smooth muscle cell proliferation, and angiogenesis [6].

Ponatinib, on the other hand, has been shown to increase in vitro levels of desmosterol, as well as other substrates of 24-dehydro-cholesterol reductase (DHCR24), which is responsible for converting desmosterol into cholesterol [59]. Desmosterol appears to possess a physiological role in vivo by modulating cholesterol metabolism and the inflammatory responses in foam cells, and its depletion in myeloid cells via the overexpression of DHCR24 seems to favor the progression of atherosclerosis by attenuating macrophage anti-inflammatory mediators [60–62]. In addition, a study conducted on APOE*3Leiden.CEPT transgenic mice reported reduced plasma levels of VLDL-LDL and cholesterol in the liver [63].

### 2.6. Renal Impairment

Renal dysfunction associated with TKIs, especially imatinib and bosutinib [64,65], can further increase cardiovascular risk [66]. Pieces of evidence in the literature have reported this association and several hypotheses have been advanced.

Renal injury may result from a tumor lysis syndrome [67], while vomiting and diarrhea, commonly known side effects of bosutinib and imatinib, may all contribute to the hypovolemic state that could induce renal dysfunction [68]. Direct toxic tubular damage can depend on imatinib's ability to inhibit the PDGFR, which usually regulates the proliferation and regeneration of tubular cells [66]. Increased serum creatinine may also depend on the inhibitory action of imatinib and bosutinib toward the Organic Cation Transporter 2 (OCT2), encoded by the SCL22A2 gene [69,70]. In this regard, a 20% increase in serum creatinine was observed with a median serum bosutinib concentration of 60–70 ng/mL, with a significant increase, especially in patients harboring the 808G/G polymorphism of SCL22A2 [70].

Cases of acute kidney injury (AKI) have been reported in patients treated with dasatinib. Ozkurt et al. reported a case of gastroenteritis and acute kidney injury in a patient with imatinib-resistant CML, which was resolved after the discontinuation of dasatinib [71]. Additionally, case reports have described the development of nephrotic syndrome in adult and pediatric CML patients. Dasatinib, being a multi-kinase inhibitor, may potentially damage podocyte and endothelial cells through the direct inhibition of SRC kinases, with indirect effects on VEGF [72].

Nevertheless, these mechanisms remain unclear, and other factors, alone or in combination, may contribute to kidney damage. Further investigations are needed to clarify the mechanisms underlying renal impairment during imatinib treatment and to identify potential predictive factors.

### 2.7. Platelet Dysfunction

Ischemic heart disease and stroke are one of the leading causes of mortality globally. The underlying cause of these conditions is often thrombotic complications resulting from the rupture or erosion of atherosclerotic plaques. Antiplatelet therapy (e.g., aspirin and clopidogrel) is the mainstay of the current treatment for arterial thrombosis, but it has limitations, such as irreversible action and increased bleeding risk [14].

Platelet dysfunction has been reported in 40% of patients treated with dasatinib, with 10% of them experiencing grade ≥3 dysfunction. Specifically, the drug appears to impair platelet aggregation, without being associated with bleeding diathesis in treated patients. Quintas-Cardama et al. demonstrated that dasatinib did not interfere with secondary hemostasis, but rather affected platelet aggregation, similar to aspirin. The authors examined this platelet aggregation in 87 patients with CP-CML, including 27 patients treated with dasatinib, 32 with bosutinib, 19 with imatinib, and 9 with nilotinib. Among the patients treated with dasatinib, impaired platelet aggregation upon stimulation with arachidonic acid, epinephrine, or both was observed in 70%, 85%, and 59% of them, respectively [73]. A second study evaluating the correlation between TKI-induced platelet

dysfunction and bleeding diathesis in CML patients found no correlation between bleeding presence, bleeding scores, and platelet dysfunction [74].

Dasatinib interferes with platelet aggregation, which is important because antiplatelet agents such as aspirin or clopidogrel are commonly used to prevent thrombotic cardiovascular events. While caution is advised when combining anticoagulant and antiplatelet therapy with dasatinib, especially in elderly patients or those prone to bleeding, the overall risk of bleeding without these factors is likely low [75].

## 3. Cardiovascular Toxicity Risk Assessment and Stratification

In the "TKIs era", the need to evaluate the cardiovascular risk, both in the initial treatment decision and in the long-term management of CML, has become established. The identification and management of risk factors are invaluable for recognizing the patients most likely to develop CTR-CVTs and implementing strategies to mitigate or reverse the risk at baseline. For this purpose, the first Guidelines on Cardio-Oncology were published in 2022 by the European Society of Cardiology (ESC), providing a personalized approach to care based on basal cardiovascular toxicity risk assessment and new surveillance protocols during cancer treatment [14].

The absolute risk for CTR-CVTs is correlated with cardiovascular risk at baseline and changes over time during treatment. Therefore, diagnosis and before starting TKIs are the optimal timepoints for the elaboration of preventive strategies. History assessment, physical examination, blood pressure (BP) measurement, and glucose and lipids dosing are essential to stratifying CML patients within a specific risk category and achieving a considered therapeutic decision. Other cardiac tests (i.e., ECG, biomarkers, and imaging) should be considered individually based on the cardiovascular risk and planned treatment strategy (Figure 2). Especially with nilotinib or ponatinib, cardiovascular risk assessment is recommended every 3 months during the first year and every 6–12 months thereafter [14].

In recent years, the most widely used CV risk assessment system in the general population is the Systemic Coronary Risk Estimation (SCORE) algorithm, which evaluates the 10-year risk of death for CVD based on sex, age, systolic pressure, smoking, and total cholesterol levels [76]. In CML, the SCORE algorithm was applied retrospectively in 85 patients treated with ponatinib, where patients with a high to very high SCORE risk showed significantly higher rates of AOEs than those with lower risk classes (74.3% vs. 15.2%, $p < 0.001$) [77]. The same algorithm, when applied to a cohort of 369 patients treated with nilotinib, showed a significantly higher incidence of AOEs (34.4 $\pm$ 6% vs. 10 $\pm$ 2.1%, $p < 0.001$) in patients in a high and very high SCORE risk class (SCORE > 5%) [78]. Overall, these data appear to be in line with those reported in a recent meta-analysis, where high and very high SCORE risks were correlated with a higher risk for AOEs in CML patients treated with nilotinib (HR 3.5 IV 95% 1.4–8.7 and HR 4.4 IC 95% 2–9.8, respectively) [6].

In 2021, the SCORE algorithm was updated in the SCORE2 for apparently healthy people of 40–69, with the inclusion of fatal and non-fatal events (e.g., myocardial infarction and stroke), and SCORE2-Older Patients (SCORE2-OP) for those aged >70 years [14]; compared to the previous one, the updated algorithm (1) has been validated on much larger derivation and validation cohorts, (2) provides risk estimates for the combined outcome of 10 years of fatal and non-fatal CVD events, and (3) improves the overall risk discrimination, especially in younger patients [79]. However, this updated version has considerable advantages over the previous one, but still needs to be evaluated on CML patients.

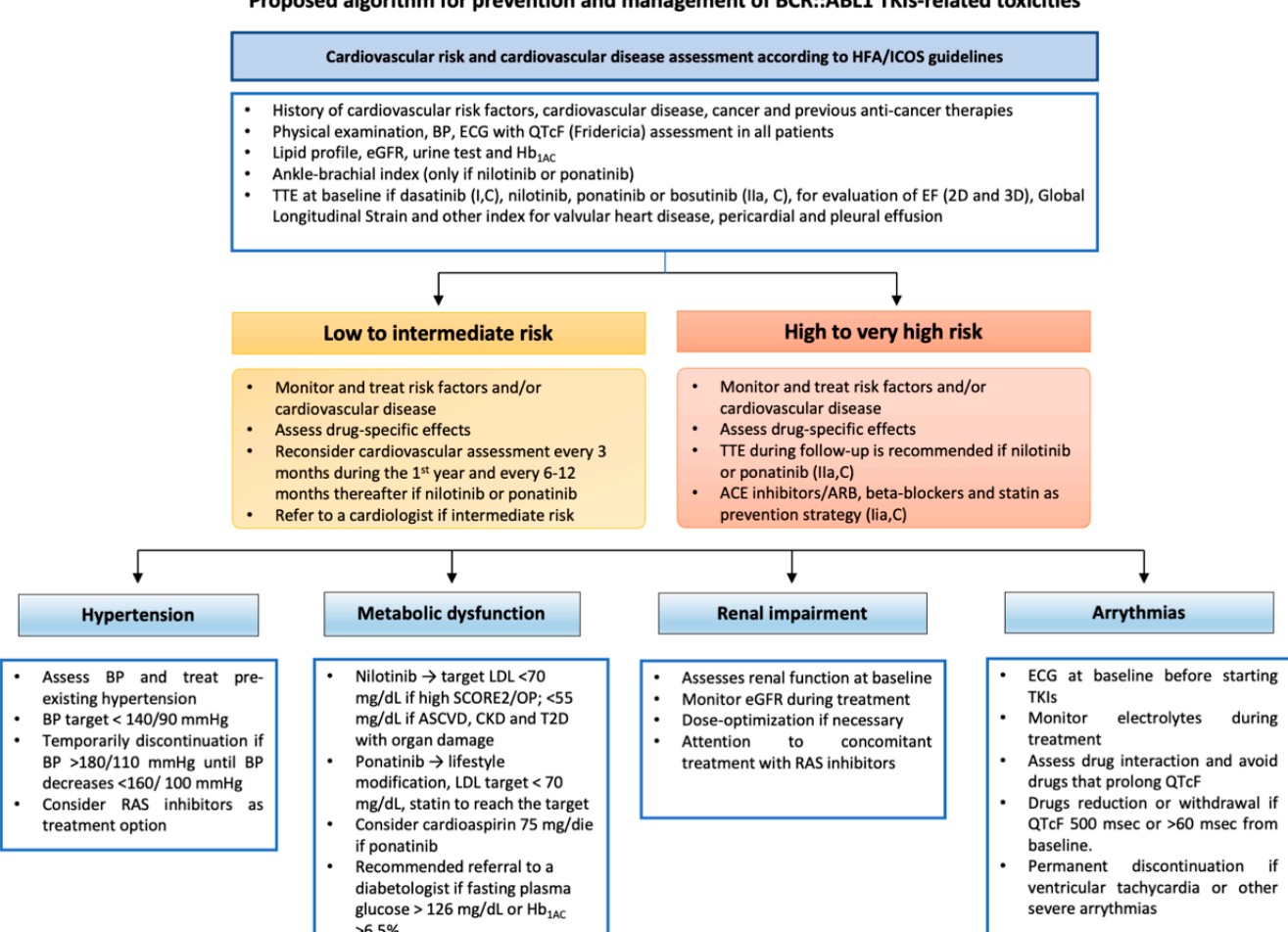

**Figure 2.** Proposed algorithm for prevention and management of BCR::ABL1 TKIs cardiovascular toxicities according to the ESC Guidelines on CVR prevention and the most recent ESC guidelines on Cardioncology. The algorithm emphasizes the importance of mitigating CVR after adequate assessment of risk factor at baseline according to the new HFA/ICOS scoring system. With this aim, lifestyle intervention, implementation of cardiovascular therapy, tailored follow-up and TKIs dose-optimization strategies are crucial. Recommendation classes and levels of evidence are also reported where possible. ACE inhibitors: Angiotensin Converting Enzyme inhibitors; ARB: Angiotensin Receptor Blockers; ASCVD: Atherosclerotic Cardiovascular Disease; BP: Blood Pressure; CKD: Chronic Kidney Disease; EF: Ejection Fraction; eGFR: estimated Glomerular Filtration Rate; LDL: Low-density Lipoprotein; RAS inhibitors: Renin-Angiotensin System Inhibitors; SCORE2/OP: Systemic Coronary Risk Estimation/SCORE2 Older people; Tyrosine Kinase Inhibitors; TTE: transthoracic Echocardiogram; and T2D: Type 2 Diabetes.

The recently proposed HFA/ICOS risk assessment tool provides a new stratification tool at baseline in cancer patients undergoing cardiotoxic therapies, according to the category of exposure drugs, including BCR::ABL1 TKIs, and based on cardiovascular risk factors [15]. This score, included in the ESC Guidelines on Cardio-oncology, was retrospectively validated on a monocentric cohort of 58 CML patients, 32 of whom were men, with median age of 59 ± 15 y.o., where the HFA/ICOS stratification model allowed for a better risk evaluation of CML patients with a higher sensitivity when compared to the SCORE algorithm [80].

Regardless of the risk category, some general measures should be pursued, such as encouraging a healthy lifestyle, stopping smoking, and following a healthy and balanced diet. Moreover, risk modifiers (i.e., psychosocial factors, ethnicity, and imaging), lifetime

CVD risk, treatment benefits, comorbidities, frailty, and patient preferences should also be taken into account.

Once stratified, if significant cardiovascular risks are found, it is mandatory to report the patients to a cardiologist in order to carry out further investigations and correct them before starting CML-specific therapy [14].

### 3.1. Modifiable Risk Factors

### 3.1.1. Hypertension

Arterial hypertension is one of the most important causes of preventable morbidity and mortality worldwide, affecting about 30–45% of the general population and predisposing them for severe pathologies such as coronary artery disease (CAD), cerebrovascular disease, atrial fibrillation (AF), and chronic kidney disease (CKD) [81]. In addition, it is also the most commonly reported cardiovascular comorbidity in cancer registries, reaching up to a third of cancer treatment [82].

In the CML population, the rate of hypertension was 10% in patients treated with 2G/3G-TKIs, especially in association with nilotinib (RR 2, $p$ = 0.0002) and ponatinib (RR 9.21, $p$ = 0.0002) [83]. In the last 10-year update of the pivotal ENESTnd trial, 16.1% of patients in the nilotinib 300 mg BID arm vs. 20.2% in the nilotinib 400 mg BID arm vs. 6.1% in the imatinib 400 mg QD arm developed hypertension [84]. In a real-life cohort, 94 patients were treated with frontline nilotinib, with a median age of 58 years at diagnosis, and arterial hypertension was the most commonly reported comorbidity (23.4%); after 28.6 months of treatment, 6 patients developed CTR-CVTs, 3 of which had baseline hypertension [85].

A high BP has also been found in several trials with ponatinib, both in pivotal trials and in real-world experiences. Notably, in 37% of the patients included in the PACE trial and 31% in the phase III EPIC trial, any grade of hypertension was found [3,86]. More recently, a dose optimization strategy was investigated in the OPTIC trial, where a dose reduction in ponatinib was performed according to molecular response. More than 20% of the 283 enrolled patients had controlled hypertension at baseline. The most common non-hematological side effect was, as expected, hypertension (28%), 9% of which was grade $\geq 3$ in the last 3-year follow-up [87,88]. Finally, the OITI trial, a prospective–retrospective non-interventional trial based on the real-world data of patients treated with ponatinib, enrolled 119 patients, including 110 with CML in the Chronic Phase (CML-CP), with a median age of 60 years. Of these, 47.1% of the patients had a history of cardiovascular disease or hypertension; in a first and interim analysis, 59.7% of the patients experienced adverse events, among which grade 1–2 hypertension was one of the most commonly reported ones [89].

Among new agents, the Specifically Targeting the ABL Myristoyl Pocket (STAMP) inhibitor asciminib has been found to be particularly effective, even in presently pretreated patients, with an excellent toxicity profile. The safety and efficacy of asciminib in monotherapy or in combination with nilotinib, imatinib, or dasatinib were investigated in a phase I trial on CML patients resistant or intolerant to $\geq 2$ different TKIs. Few AEs were reported, mainly represented by an increased lipase level in 26.7%, fatigue in 29.3%, headache in 28%, hypertension in 19%, and arthralgia and nausea in 24% [90]. In the phase 3 trial ASCEMBL, CML-CP patients treated with $\geq 2$ previous TKIs were randomized to asciminib 40 mg BID vs. bosutinib 500 mg QD. At a median follow-up of 2.3 years, asciminib showed and maintained a higher efficacy than bosutinib, with a better toxicity profile. Overall, any-grade AEs were found in 91% of the patients treated with asciminib, 56.4% of which were grade 3. Among these, 18 (11.54%) experienced hypertension, of which 13 cases were within the first 12 months of treatment [91].

The pharmacological agents currently approved for hypertension treatment are divided into five classes of drugs, including Angiotensin-converting enzyme inhibitors (ACEi), Angiotensin II Receptor blockers (ARB), Beta-blockers, Calcium antagonists, and diuretics [92]. An over-activation of the RAS components in CML patients has been noted [93]. In addition, a study conducted on a large cohort of CML patients with arterial hypertension

at diagnosis and treated with 2G/3G-TKIs reported a lower incidence of AOEs in the patients treated with RAS inhibitors compared to other treatments available (14.8% ± 4.2% vs. 44 ± 1%, *p* < 0.001) [94]. Overall, these data align with what is already known about the pro-atherogenic role of over-activated RAS in arterial hypertension and suggest the potential role of RAS inhibitors in CVD prevention for CML patients.

Recommendations: based on the aforementioned evidence, nilotinib and ponatinib have been shown to increase BP levels, so attention should be paid to patients with baseline hypertension or other cardiovascular risks factors, such as a history of CVD, T2D, or kidney failure. Standardized BP measurements, target organ damage research, and the exclusion of secondary causes of hypertension are necessary before TKI treatment. According to the evidence available in the literature, RAS inhibitors (ACEi or ARBs) could be a good option compared to other anti-hypertensive classes in patients with CML and treated with 2G/3G-TKIs TKIs [93].

### 3.1.2. Dyslipidemia

Dyslipidemia has a causal role in atherogenesis and is therefore considered to be one of the major cardiovascular risk factors: increased plasma levels of LDL can cause endothelial accumulation, thus inducing dysfunction and the subsequent inflammation of the vascular wall [95].

Treatment with nilotinib has been associated with dyslipidemia. Rea et al. first reported increased triglycerides and cholesterol plasma levels in 27 CML patients treated with nilotinib, identifying age, the duration of treatment, and pre-existing metabolic risk as risk factors for hypercholesterolemia [96]. Data from a large retrospective cohort of 369 CML patients treated with nilotinib showed a significantly increased incidence of AOEs at three months in patients with plasma cholesterol levels of >200 mg/dL and LDL levels of >70 mg/dL (21.9% vs. 6.2%, *p* < 0.01) [78].

Ponatinib has been proven to be particularly effective at inducing deep, early, and long-lasting molecular response, but this recourse has been limited by the increased incidence of AOEs in treated patients. Clinical trials and real-life cohorts have reported dyslipidemia as a frequent baseline comorbidity in patients who experience AOEs during treatment with ponatinib [3,86,97–99]. In particular, the association between dyslipidemia and AOEs during treatment with ponatinib was investigated in a cohort of 116 CML patients. Overall, 22% of the patients had dyslipidemia at diagnosis with CML, and 35% before starting ponatinib; a cardiovascular risk assessment using the SCORE algorithm defined 72% of patients as low risk (SCORE ≤5%) for CVD, while 28% were defined as high to very high risk (SCORE >5%). In a multivariate analysis, a high to very high SCORE risk confirmed this significant association with AOEs (HR 2.9, IC 95% 1–9.1; *p* = 0.04). In addition, a higher incidence of vascular events was reported in patients with pretreatment triglyceride levels of >200 mg/dL and after three months of treatment in patients with plasma cholesterol levels of >200 mg/dL and LDL levels of >70 mg/dL [100].

Recommendations: Nilotinib or ponatinib may be associated with an increased risk of CTR-CVTs, especially in older patients or those with pre-existing CV risk factors.

According to the 2022 ESC guidelines, for a comprehensive cardiovascular risk assessment, baseline laboratory tests, including those on lipid profile (total cholesterol, LDL, HDL, and triglycerides) and glycated hemoglobin (HbA1c), are recommended prior to initiating therapy [14]. Based on specific evidence derived from the CML population and in accordance with the 2021 ESC guidelines on cardiovascular risk prevention, special attention should be given to patients with a high SCORE2/OP risk who are candidates for nilotinib. In these patients, the LDL target should be <70 mg/dL. Caution is recommended in patients with a history of atherosclerotic cardiovascular disease (ASCVD), chronic kidney disease (CKD), and type 2 diabetes (T2D) with organ damage, particularly in very-high-risk patients. In these cases, plasma cholesterol levels should be maintained below 55 mg/dL [78,101].

Based on specific evidence derived from the CML population, patients undergoing ponatinib treatment aged >60 with established ASCVD, dyslipidemia, diabetes, or another cardiovascular risk factor, should be considered to be at a high risk for CTR-CVTs [100]. Lifestyle modification and target levels of LDL of <70 mg/dL are recommended for these patients. Moreover, a lipid-lowering statin-based therapy should be suggested to keep LDL levels <70 mg/dL. A combination with ezetimibe can be considered when the maximum dosage of statin therapy alone is insufficient to reach the target value. Treatment with a statin has been associated with adverse events that may limit its efficacy in lipid lowering; in these cases, a different schedule, such as every other day or twice a week, may be used [101]. Finally, prophylaxis with aspirin at 75–100 mg/day, or with clopidogrel at 75 mg/day in case of aspirin intolerance, should be considered in patients with CV risk factors, especially if they are aged >60 or have a preexisting history of CVD [77,102].

### 3.1.3. Diabetes and Pre-Diabetes

Diabetes mellitus and pre-diabetes are known independent risk factors for CAD and cerebrovascular disease, doubling the risk of ASCVD, and are frequently concomitant to dyslipidemia and hypertension. If not properly treated, diabetes is also responsible for life-threatening complications such as CKD, neuropathy, and retinopathy. Its treatment options include lifestyle changes, diet, oral hypoglycemics, and insulin therapy [103]. The increasing prevalence of T2D makes clear the importance of adequate glucose management, even in the CML population. Although imatinib and, less clearly, dasatinib, appear to have a positive effect on glucose metabolism, nilotinib has been associated with increased glucose levels and IR status during treatment [6]. In contrast, ponatinib has not been directly associated with hyperglycemia, but AOEs occur more frequently in patients with at least one CVD risk factor at baseline, including diabetes [102].

Recommendations: Based on the aforementioned evidence, nilotinib and ponatinib should be carefully evaluated in patients with uncontrolled diabetes, and a fasting plasma glucose (FPG) assessment is recommended before starting treatment. Furthermore, referring patients to an evaluation for diabetes is recommended in the case of FPG > 126 mg/dL (≥7.0 mmol/L) or Hb1Ac > 6.5% (48 mmol/mol), and in the case of FPG 100–125 mg/dL or Hb1Ac 5.6–6.4%, which are indicative of a pre-diabetes status. According to the 2019 ESC guidelines on diabetes, smoking cessation, a healthy lifestyle, and physical activity are the cornerstones of non-drug treatment. At the same time, oral hypoglycemic medications and insulin are recommended in patients where lifestyle changes are not enough. The therapeutic targets in these patients are represented by an Hb1Ac of <7.0% [103].

### 3.1.4. Smoking

Cigarette smoking, one of the most important risk factors for cardiovascular disease in the general population, is responsible for more than 50% of all avoidable deaths in smokers and increases the risk of CVD in smokers aged <50 by about five times compared to nonsmokers [101,104,105].

Smoking has proven to be a risk factor for developing CML [106,107], but has also been associated with an extremely unfavorable prognostic significance. Lauseker et al. assessed the impact of smoking on survival and progression towards the advanced stages of the disease: the 8-year survival probability for a nonsmoking patient was 87% vs. 83% for a patient who smoked, with a risk of death that was 2.08 times higher for smokers vs. nonsmokers, and a 2.11-times higher cause-specific hazard of disease progression [108]. Indeed, cigarette smoking is notoriously associated with an increased risk of atherogenesis, to which it contributes through several mechanisms, including the reduced bioavailability of NO, an increased expression of adhesion molecules, endothelial dysfunction, and increased levels of oxidized LDL, all factors that are potentially aggravated by TKIs such nilotinib or ponatinib [109,110]. Although there are no specific studies assessing the impact of early smoking cessation after diagnosis on the prognosis of CML patients, it would be

of particular interest to assess whether the prognosis of early quitters is similar to that of nonsmokers.

Recommendations: According to the 2021 ESC guidelines on cardiovascular risk prevention, permanent smoking cessation should be encouraged in all smokers at diagnosis, since it is probably the most effective of all the preventive measures. Drug support includes nicotine replacement therapy (chewing gum, transdermal nicotine patches, nasal spray, inhaler, and sublingual tablets), bupropion, varenicline, and cytisine, although specific studies on the CML population are lacking [101].

### 3.1.5. Obesity

Obesity is a common disorder associated with an increased risk of comorbidity, including CVD, CKD, T2D, and hypertension [105], and many studies have supported the association between a high Body Mass Index (BMI) and an increased risk of cancer, both solid and hematologic [107]. Moreover, obesity has been recognized as a risk factor for CML [111,112].

The outcomes of obese CML patients were investigated by Breccia et al., who reported a correlation between an increased BMI and response to imatinib vs. nilotinib and dasatinib, with a delay in achieving a Complete Cytogenetic Response (CCyR, 6.8 months vs. 3.3 months, $p = 0.01$), a reduced incidence of major molecular response (MMR, 77% vs. 58% $p = 0.03$), and an increased rate of resistance [113]. In contrast, subsequent studies have found no significant difference in the molecular responses between patients with a low (<25 kg/m$^2$) and high (>25 kg/m$^2$) BMI treated with nilotinib or dasatinib in the front line [114].

Among the therapies for obese patients with a BMI $\geq$ 35 mg/m$^2$ is bariatric surgery [115]. These surgeries alter bioavailability and drug metabolism, so these patients are often excluded from clinical trials. In a retrospective analysis of 22 patients with a history of bariatric surgery, a lower rate of early molecular responses (68% vs. 91%; $p = 0.05$) and a longer time to CCyR (6 vs. 3 months; $p = 0.001$) or MMR (12 vs. 6 months; $p = 0.001$) were reported when compared to 44 matched-control patients with no history of bariatric surgery. Moreover, bariatric surgery was associated with inferior event-free survival (5-year, 60% vs. 77%; $p = 0.004$) and failure-free survival (5-year, 32% vs. 63%; $p < 0.0001$) [116].

Recommendations: Based on specific evidence derived from the CML population and in accordance with the 2021 ESC guidelines on cardiovascular risk prevention, BMI should be included in the preliminary evaluation before starting TKI treatment. Compared to imatinib, 2G-TKIs appear to be the most suitable choice in obese subjects who need to achieve early and deep molecular responses, although cardiovascular risk and other comorbidities should nevertheless be considered in this subgroup of patients [113,114]. Attention should be paid, finally, to obese patients eligible for bariatric surgery, since it can alter TKIs' absorption and metabolism, being able to influence patient outcomes. Despite this, no data support a change of dose in patients with a history of obesity or candidates for bariatric surgery.

### 3.1.6. Renal Function Impairment

Chronic Kidney Disease (CKD), defined as abnormalities in the structure or kidney function that persist for 3 months [117], is an important cardiovascular risk factor, with its mortality progressively increasing to triple when the estimated Glomerular Filtration Rate (eGFR) reaches 15 mL/min/1.73 m$^2$ [101].

In CML, several studies in the literature have reported impaired glomerular function, especially in patients treated with imatinib. A median eGFR reduction of 2.77 mL/min/1.73 m$^2$/years ($p < 0.001$) was reported in a cohort of 105 CML patients treated with imatinib, 12% of which developed CKD and 7% developed AKI [118]. In a retrospective analysis of 397 CML patients, 320 of them were treated with imatinib, 25 with dasatinib, and 53 with nilotinib. In the imatinib group, a median reduction in eGFR of 3 mL/min after 12 months of treatment was found. In addition, the authors reported an increased rate of

CTR-CVTs in the imatinib cohort when compared to patients treated with 2G-TKIs [119]. A recent meta-analysis confirmed these data, reporting a significantly higher risk of renal impairment (CKD or AKI) in patients treated with imatinib than other TKIs (RR 2.7 IC 95% 1.49–4.91) [64].

An improvement in renal function was reported in 60 patients previously treated with imatinib that switched to a next-generation TKI [120], but nilotinib and dasatinib in the second- or third-line still seem to have mild effect on kidney function [66].

Nephrotoxicity was also explored by Cortes et al. in patients receiving bosutinib, both in the front-line and after one or more TKI. Like imatinib, bosutinib was associated with a median eGFR reduction of $-15.62$ mL/min/1.73 m$^2$ from baseline to 48 months, with the use of loop diuretics, aminoglycosides, anti-hypertensive drugs, and grade 3–4 diarrhea being time-dependent prognostic factors for an eGFR of $< 45$ mL/min/1.73 m$^2$ [65]. To note, renal dysfunction during the simultaneous intake of imatinib or bosutinib and anti-hypertensive therapy was reported in another experience, with rates of eGFR reduction of $-10.1\%$/year ($-12.3, -7.9$; $p = 0.04$) and $-5.7\%$/year ($-6.6; -4.9$; $p < 0.01$), especially when treated with RASi [68].

Recommendations: Based on the aforementioned evidence, TKIs have been shown to be safe when administered at standard doses in patients with modest levels of renal dysfunction. Imatinib and, to a lesser extent, bosutinib, have been associated with renal dysfunction in long-term management [64,65]. According to expert opinions, patients can be treated with the most suitable TKI and subjected to careful monitoring of their kidney function, with dose optimization if necessary. Moreover, the eGFR should be carefully monitored in patients simultaneously receiving RASi and imatinib or bosutinib [68].

### 3.2. Non-Modifiable Risk Factors

### 3.2.1. Age and Frailty

Age is considered to be a major driver of CVD risk, as incidence increases with ageing, both in men and women, exceeding 75% in people between 60 and 79 years and more than 86% in those aged >80 years. The high prevalence of CVD in older people has been linked to increased oxidative stress, inflammation, apoptosis, and myocardial dysfunction and deterioration [101,121].

The median age for CML diagnosis is between 50 and 60 years, with about 50% of cases being diagnosed in people aged $\geq 60$ years [122]. Advanced age in CML patients can profoundly impact their management, as it can be a problem for the progressive deterioration of performance, co-morbidity, polypharmacy, and outcomes [123]. Finally, frailty, defined as a reduced resistance of the individual to stressors, is crucial to consider at diagnosis before starting therapy. It is also important to include functional capacity, co-morbidity burden, level of social support, and nutritional status [124]. In a retrospective analysis of 810 elderly patients with CML, with a median age of 75 (range 70–80), 26% had an underlying CVD and 86% had at least one CV risk factor. In the front-line, 63.1% received imatinib, 23.1% dasatinib, 12.5% nilotinib, and less than 1.4% bosutinib or ponatinib. Overall, 65.3% and 21.1% of patients received one or two lines of therapy, respectively. A multivariate analysis showed that the first-line patients treated with dasatinib were more likely to switch to a second line than those with imatinib (HR 1.9, IC 95% 1.35–2.66, $p < 0.01$). In addition, patients with more comorbidities were less likely to change from a first-line TKI than those without comorbidity (HR 0.66, IC 95% 0.46–0.95, $p = 0.02$) [125].

Recommendations: Based on specific evidence derived from the CML population and expert opinions, starting treatment with TKIs is crucial even in elderly patients, as they were all proved effective when elderly patients were compared to younger patients. Nevertheless, when selecting TKI therapy, special attention should be paid to comorbidities, polypharmacy, and frailty, since they can affect survival. Thus, imatinib seems to be the best choice to date for initial therapy in elderly patients with CML, due to its good cardiovascular toxicity profile. Dasatinib should be avoided, where possible, in patients with underlying lung disorders (e.g., Chronic Obstructive Pulmonary Disease or Pulmonary

Hypertension), while nilotinib and ponatinib should be avoided in elderly patients with CVD risk factors [126].

### 3.2.2. Gender

The current international guidelines recommend integrating sex, gender, and gender identity considerations into individuals' risk assessment and clinical management. In fact, not only do women present unique CVD risk factors (e.g., Polycystic Ovaric Syndrome or pregnancy-associated conditions), but seem to present different CVD manifestations and different responses to treatments [127].

Studies after introducing TKIs have shown no gender differences in the overall survival or response to TKIs. However, few studies have investigated the role of gender in CTR-CVTs during TKI therapy. Among them, a case–control study of AEs recorded in the FDA Adverse Events Reporting System (FAERS) database found that the CTR-CVTs rate in CML was higher in men than women exposed to TKIs (52.9% vs. 36.2%), compared to patients exposed to other anticancer drugs [128]. Karantanos et al., investigating the biological mechanism behind sex-related differences in Chronic Myeloid Neoplasm, including CML, reported a higher mutation frequency in DNMT3A and TET2 in women, which are known to be correlated with a higher CVD risk, but also a higher incidence in men of non-specific-MPN mutations, especially in ASXL1, IDH1/2, U2AF1, SRSF2, and EZH2, which are associated with worse outcomes in MPN patients [129].

Recommendations: Currently, there is no evidence to suggest a different CTR-CVTs risk prevention gender-based approach in the CML population. Rather, considering the data presented, it would be interesting to carry out an ad hoc study to assess the impact of gender on the risk of CTR-CVTs in patients treated with TKIs.

## 4. Conclusions

CTR-CVTs represent one of the main issues related to TKI treatment, thus limiting the available therapeutic options and leading to a dramatic worsening of prognoses. Moreover, CTR-CVTs have a strong impact on mortality, independent from CML. The current manuscript highlights the importance of multidisciplinary evaluations in CML patients, balancing the need to pursue TKI therapy with the specific burden of CVD and its risk. Finally, we emphasize tailored cardiovascular risk stratification in every CML patient. Accurate prevention protocols, the availability of safer agents, and dose optimization strategies may be the key to reducing the mobility and mortality in these patients.

**Author Contributions:** Conceptualization, A.C. and O.M.; writing—original draft preparation, A.C., R.P. and O.M.; writing—review and editing, A.C., R.P., O.M., G.C., F.M., F.T. and G.L.N.; supervision, G.L.N. All authors have read and agreed to the published version of the manuscript.

**Funding:** This research received no external funding.

**Institutional Review Board Statement:** Not applicable.

**Informed Consent Statement:** Not applicable.

**Data Availability Statement:** No new data were created or analyzed in this study. Data sharing is not applicable to this article.

**Conflicts of Interest:** The authors declare no conflict of interest.

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
