# Peer review of "The Direct and Indirect Effects of Tyrosine Kinase Inhibitors on the Cardiovascular System in Chronic Myeloid Leukemia"

_hemato, doi:10.3390/hemato4030017_

Round 1
Reviewer 1 Report
In this manuscript, Costa et al. gave an extensive and inclusive review of recent reports on the therapy-related cardiovascular toxicities (CVTs) associated with tyrosine kinase inhibitors therapy against CML. As the authors wrote, it is now crucial to evaluate possible CVTs in advance and implement prevention protocols. Thus, this manuscript is timely and well-written in general. The followings are the points which should be addressed.
1. Section 2.3 Arrythmias. It seems questionable that dasatinib actually cause ventricular arrhythmias via their inhibitory effects on hERG, because dasatinib showed that activity with IC50 of 14.3μwhile its toxic effect on CML cells is <IC50 of 1 nM.
2. Section 2.6 Renal impairment. As the authors mentioned, its precise mechanism is still unclear. Although the authors emphasize the renal toxicity of imatinib and bosutinib, other TKI such as dasatinib has been reported to result in renal damage. BMC Nephrology (2019) 20:87; Ren Fail (2010) 32(1):147-9.
Author Response
Dear Reviewer,
Thank you for your feedback on our article.
- We appreciate your concerns regarding the potential of dasatinib to cause ventricular arrhythmias through its inhibitory effects on hERG, given its demonstrated activity with an IC50 of 14.3 μ We acknowledge that despite this activity, dasatinib has shown a toxic effect on CML cells with an IC50 value below 1 nM, suggesting a primary focus on tumor cells rather than cardiac cells. Considering this, we agree that it may be debatable whether dasatinib effectively causes ventricular arrhythmias solely through its effects on hERG. It is worth noting that individual variations and additional factors, such as drug interactions and genetic predisposition, could contribute to the overall risk. We acknowledge the need for further studies and comprehensive evaluations to better understand the potential relationship between dasatinib, hERG inhibition, and the occurrence of arrhythmias. These investigations will provide valuable insights into the specific mechanisms and risks associated with dasatinib treatment.
- Additionally, we have further explored the topic of dasatinib-induced renal damage using the available data and the bibliography you kindly provided.
Once again, we appreciate your valuable input and will take your comments into account as we continue to refine our research.
Reviewer 2 Report
In this excellent review, the authors succeeded in covering the subject in a comprehensive manner without being redundant. The structure of the article clearly separates pathophysiologic mechanisms from clinical effects and recommendations. The following points should be revised:
1) In the second part of the review, the authors emit recommendations on baseline and follow-up investigations, choice of TKI and supportive care modalities. The methodology of those recommendations should be clarified: based on the literature, expert opinion, already established recommendations, other?
2) Dasatinib and increased risk of bleeding should also be mentioned, especially in situations of cardiovascular treatment with aspirin.
3) The two figures are difficult to read because of the small size of the text and the multitude of abbreviations, which might be often used by cardiologists but not by hematologists. Less abbreviations should be used.
Author Response
Dear Reviewer,
Thank you for your valuable feedback on our article. We have addressed your comments and made the following updates:
1) Regarding the recommendations in the second part of the review, we have clarified that they are based on a comprehensive analysis of the existing literature, expert opinions, and already established recommendations in the field. We have provided additional information to better highlight the methodology used in formulating these recommendations.
2) We have included a paragraph discussing dasatinib and its association with an increased risk of bleeding, particularly in situations involving cardiovascular treatment with aspirin. We emphasize the need for caution when administering dasatinib concurrently with antiplatelet agents, especially in patients with bleeding predispositions or in elderly patients.
3) We have taken your suggestions into consideration and made efforts to reduce the use of abbreviations pictures added to the paper. We have also enlarged the text where possible to improve readability. However, due to limitations in modifying the images, we are unable to make further modifications without compromising the graphics and the message they convey.
Thank you once again for your valuable input. We have incorporated your suggestions to enhance the clarity and comprehensibility of our article.